# Interpreting T-cell search "strategies" in the light of evolution under constraints

**Inge M. N. Wortel** [1,2]*, **Johannes Textor**[1,2]

**1** Medical BioSciences, Radboudumc, Nijmegen, the Netherlands, **2** Data Science, Institute for Computing and Information Sciences, Radboud University, Nijmegen, the Netherlands

* inge.wortel@ru.nl

**Data Availability Statement:** All code required to reproduce this manuscript is available on Github (https://github.com/ingewortel/2022-Tcell-evolution, archived at https://doi.org/10.5281/

## Abstract

Two decades of *in vivo* imaging have revealed how diverse T-cell motion patterns can be. Such recordings have sparked the notion of search "strategies": T cells may have evolved ways to search for antigen efficiently depending on the task at hand. Mathematical models have indeed confirmed that several observed T-cell migration patterns resemble a theoretical optimum; for example, frequent turning, stop-and-go motion, or alternating short and long motile runs have all been interpreted as deliberately tuned behaviours, optimising the cell's chance of finding antigen. But the same behaviours could also arise simply because T cells *cannot* follow a straight, regular path through the tight spaces they navigate. Even if T cells do follow a theoretically optimal pattern, the question remains: which parts of that pattern have truly been evolved for search, and which merely reflect constraints from the cell's migration machinery and surroundings?

We here employ an approach from the field of evolutionary biology to examine how cells might evolve search strategies under realistic constraints. Using a cellular Potts model (CPM), where motion arises from intracellular dynamics interacting with cell shape and a constraining environment, we simulate evolutionary optimization of a simple task: explore as much area as possible. We find that our simulated cells indeed evolve their motility patterns. But the evolved behaviors are not shaped solely by what is functionally optimal; importantly, they also reflect mechanistic constraints. Cells in our model evolve several motility characteristics previously attributed to search optimisation—even though these features are *not* beneficial for the task given here. Our results stress that search patterns may evolve for other reasons than being "optimal". In part, they may be the inevitable side effects of interactions between cell shape, intracellular dynamics, and the diverse environments T cells face *in vivo*.

## Author summary

T cells are immune cells with an astounding ability to move nearly anywhere in the body. This motion helps them detect and clear up pathogens, and understanding it is key to understanding T-cell immunity.

zenodo.7635033). This repository also contains a website with an interactive simulation and S1 Movie, accessible through: https://ingewortel. github.io/2022-Tcell-evolution/.

**Funding:** This work was funded by a Human Frontiers Science Program (HFSP, https://www. hfsp.org/) grant RGP0053/2020 awarded to JT (IMNW was also supported by this grant). In addition, IMNW was supported by a Radboudumc PhD grant, and JT by a Vidi grant VI.Vidi.192.084 from the Dutch Research Council (NWO, https:// www.nwo.nl/en). The funders had no role in study design, data collection and analysis, decision to publish, or preparation of the manuscript.

**Competing interests:** The authors have declared that no competing interests exist.

Importantly, the continuous search for pathogens means that T cells face different challenges throughout their lifetime: their needle-in-a-haystack quest for the first signs of disease in lymph nodes differs greatly from their motion in an infected lung, or from how they patrol the skin to guard against future reinfections. These observations have raised the intriguing question: have years of evolution equipped T cells with distinct search "strategies", optimized for whichever searching tasks they might encounter?

Although several studies have addressed this question in mathematical models, to date, none have explicitly considered the evolutionary process itself. Here, we directly simulate evolutionary optimization of T-cell search. We find that explicitly simulating "survival of the fittest searchers" can shed new light on why T cells move the way they do. Importantly, we find that the evolving movement patterns are only in part optimized "strategies"— while other parts may merely be "side effects" stemming from the constraints arising from the cell's molecular motor acting in a maze-like environment.

## Introduction

T cells have the rare ability to migrate in nearly all tissues within the human body. In lymphoid organs, such as the thymus and lymph nodes, T cells must migrate to develop and get activated; in peripheral "barrier" tissues, like the lung, the gut, and the skin, T cells continuously patrol in search of foreign invaders. Although T cells stay motile in these different contexts, they do adapt their morphology and migratory behaviour to environmental cues. Naive T cells rapidly crawl along a network of stromal cells in the lymph node, alternating between short intervals of persistent movement and random changes in direction [1–4]. This "stop-and-go" behaviour lets them cover large areas of the lymph node quickly, and seems to be a good strategy for finding rare antigens without prior information on their location [5–8]. Developing T cells adopt a similar strategy to find their specific ligand during negative selection in the thymic medulla [9, 10]. By contrast, positive selection in the thymic cortex involves migration at much lower speeds—perhaps due to the broader distribution of positively selecting ligands in the thymus [11]. This remarkably flexible behaviour has been suggested to reflect different "search strategies", whereby T cells maximise their chance of finding antigen [5].

The idea of search strategies has interesting implications. If T-cell migration patterns are indeed optimised for some specific function (or several functions depending on context and environment), then comparing their "search efficiency" can help us make sense of how T-cell function relates to these diverse migratory behaviours [12]. However, two problems currently limit the conclusions we can draw from this work.

The first problem is that such optimality reasoning hinges on a tacit but crucial assumption: that we observe a given behaviour because it is somehow optimal and has been selected through evolution. Yet it is well known by evolutionary biologists that evolving "optimal" behaviours through natural selection is not as trivial as it appears at first glance [13]. For example, there is an indirect mapping between *genotype* (the genes controlling cell migration that can be transferred to the next generation) and the resulting *phenotype* (observed migratory behaviour). In other contexts, such indirect genotype-phenotype mappings have been shown to affect the behaviours that can evolve through natural selection in interesting and non-trivial ways [14, 15]—yet this topic has received little attention in the context of T-cell search so far.

It is true that cells probably *can* control migratory traits such as "speed" and "persistence", at least to some extent, by evolving their genetic background or gene expression. But if they cannot tune these traits independently, they may not be able to evolve the one without

affecting the other. Indeed, this seems to be the case: a "universal coupling between speed and persistence" (UCSP) has been described in which faster cells move more persistently [16–18]. This phenomenon is thought to apply across many types of migrating cells because actin polymerisation and polarisation are inherently coupled in any type of actomyosin-driven motion [18]. Thus, the cell's migration machinery already poses constraints on the motion patterns cells can adopt. These constraints are strengthened further by the complex, crowded environment T cells typically migrate in, which can also strongly affect T-cell shapes and migration patterns [3, 19, 20]. All of these constraints mean that (evolving) T cells can only "choose" from a limited range of motion patterns, and that their behaviour will in many cases reflect some kind of compromise rather than a true optimum. The question then becomes: how do we untangle these different influences?

The second problem is that, to determine how "optimal" a migration pattern is, we must make assumptions; after all, even though it may be very useful for a searching T cell to be in two places at once or to move at the speed of light, we typically do not consider these options in a search for optimal behaviours. Put simply: we can only assess the "efficiency" of a strategy *relative* to a set of other strategies we think the cell can adopt [13, 21]. Studies investigating immune cell search mostly use (variations of) random walk models for this purpose [5, 22–26]. These mathematical or agent-based models can produce different motility patterns depending on parameters, which directly impose properties like speed and turning behaviour on the cell. For a given dataset, fitting these parameters yields a model of the "observed" strategy whose search performance we can assess on imaginary targets *in silico*. Thus, we learn whether the observed motion pattern was a good strategy for some searching task.

Such models, however, are hard to interpret. Model selection is difficult because the same data can often be explained by multiple models depending on exactly how migration is quantified [25, 27, 28]: for example, while Harris *et al.* have claimed that T cells in the brain follow Lévy flights to find rare pathogens [22], others [29] recently cautioned that similar statistics may arise through other mechanisms. Furthermore, the search efficiency found in such models can again strongly depend on the structure of the environment [30]—and even models that differ only slightly can still make very different predictions of the area cells can explore on larger time scales [31]. But most importantly, even if these models indeed show that a behaviour benefits some T-cell function, they cannot tell us whether the same behaviour could also have arisen for another reason altogether.

To unravel which migratory patterns truly *are* optimised for search, it does therefore not suffice to construct a random walk model showing that they are beneficial in some context or other. Instead, there are other crucial points to consider—Is the proposed "optimal" strategy something a cell could realistically adopt or evolve, given the biophysical constraints of its internal migration mechanism and environment? Which migration pattern would these constraints impose on the cell if no evolutionary pressures existed? To what extent do we *need* an evolutionary explanation for the pattern in question, or could it simply be a side effect of dynamic cell motion in a complex environment?

These additional questions have received little attention in the field of T-cell search, which so far has mostly taken a "top-down" view of evolution: given an observed migration behavior, they have tested whether it could theoretically be optimal for some function [22–26]. This approach considers evolution only implicitly, as the assumed driving force behind the patterns observed. Yet evolutionary theory has benefited from a complementary "bottom-up" approach, where evolution is simulated explicitly to ask: which behaviors might we expect to evolve from known or assumed basic interactions? [14, 32]

Here, we apply this bottom-up approach to the problem of T-cell search. Examining which migration patterns emerge spontaneously from the cell's migration machinery and/or the

environment, we ask to what extent cells might still evolve or tune search strategies within those constraints. We turned to a cellular Potts model (CPM) called the Act-CPM [33], in which migration arises from a machinery where cell shape, environment, and motility interact. This model naturally captures many of the constraints acting on a migrating T cell: it reproduces the UCSP, explains how cell shape dynamics limit possible migratory patterns, and can simulate (T–)cell migration in a realistic tissue environment [34]. We now use this model to simulate an evolutionary process where cells optimise a simple task: exploring as much area as possible. Although cells do optimise their migratory behaviour for this task to some extent, they also spontaneously develop behaviours that were previously interpreted as optimal search strategies but are *not* beneficial for the task given here. We discuss what these results mean for the interpretation of T-cell migration patterns as "search strategies".

## Model

In a CPM [35, 36], cells are dynamic pixel collections that move via so-called copy attempts: by copying their identity, they can "steal" pixels from another cell at their borders (Fig 1A). These identity changes are attempted at random, but constrained by a set of rules that assign them an energetic cost $\Delta H$. As $\Delta H$ determines the success probability of each change, these energy rules ultimately govern cell behaviour in the model. For example, they can constrain a cell's size, shape, or interactions with neighbouring cells (Fig 1B). Importantly, since all cells compete for pixels on the grid through the same global energy, cells naturally interact with each other in CPMs of multicellular environments.

We have previously developed a CPM that models active cell migration based on actin dynamics [33, 34]. In the Act-CPM, pixels newly gained by a protruding cell retain their

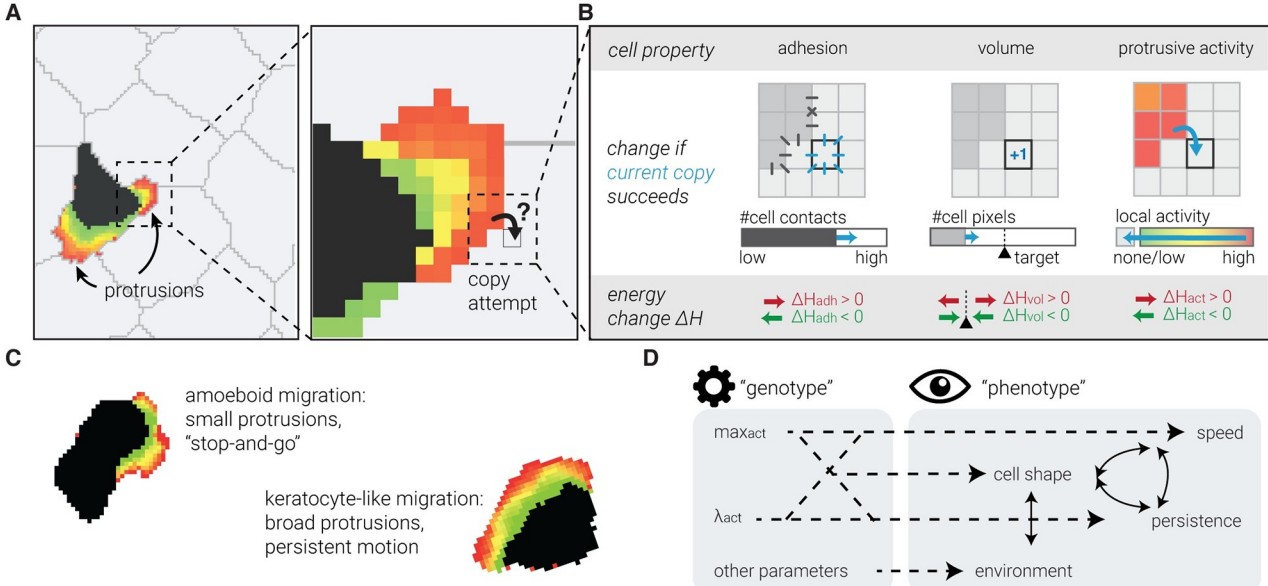

**Fig 1. A computational model of cell migration with indirect genotype-phenotype mapping.** A: CPM tissues are collections of pixels that each belong to one cell. Pixels try to copy their cell "identity" into neighbouring pixels of another cell. B: Their success rate $P_{copy}$ depends on how this would change the energy associated with different physical properties, such as surface tension ("adhesion", left), or deviating from the normal cell size ("volume", middle; an analogous constraint can be posed on the cell's perimeter). The Act-CPM [33] adds another such property (right): each pixel's "activity" represents the time since its most recent protrusive activity. Copy attempts from more into less active pixels are stimulated (negative $\Delta H_{act}$), placing positive feedback on protrusions. C: Cells in the Act-CPM can have an amoeboid (stop-and-go) or a keratocyte-like (persistent) migration mode, which are associated with different cell shapes. D: Non-trivial genotype-phenotype mapping in the Act-CPM.

"protrusive activity" for some time. When these pixels then try to copy their own identity—extending the cell with yet another pixel—their activity makes them more likely to succeed: this positive feedback gives recently active pixels a better chance of protruding again (by assigning $\Delta H_{act} < 0$; Fig 1B). Two parameters control this feedback: $max_{act}$ controls how long pixels retain their protrusive activity, while $\lambda_{act}$ tunes the protrusive strength relative to the other forces acting on the cell (i.e., those in Fig 1B). Together, $max_{act}$ and $\lambda_{act}$ control cell shape and motility. This model not only simulates cells that actively move by forming protrusions, but also reproduces different migration modes with their own protrusion shapes and motility patterns (Fig 1C) [33]. Qualitatively, it resembles the stop-and-go motility characteristic for T cells in the lymph node [1, 4].

Importantly, we have previously shown that this model also reproduces the UCSP [34]. Speed and persistence *emerge* as outputs of an intrinsic migration mechanism acting in a complex environment, rather than being imposed by the user. In evolutionary terminology, we speak of an indirect mapping from *genotype* (fixed, cell-intrinsic values of the $max_{act}$ and $\lambda_{act}$ parameters) to *phenotype* (migratory pattern), where the genotype affects the phenotype, but only indirectly (Fig 1D). Instead, interactions between the cell-intrinsic migration machinery, the cell's shape, and the structure of the surrounding tissue dynamically determine the speed and direction of motion. This essential property allowed us to examine which search behaviours (optimal or otherwise) could evolve in a system with such a non-trivial genotype-phenotype mapping.

## Results

### Act cells can evolve migratory behaviour in a simple evolutionary algorithm

We therefore simulated a simplified form of evolution by means of an evolutionary algorithm (Fig 2A, see Methods for details). We first let cells evolve to explore as much area as possible in an empty environment with no surrounding tissue (illustrated on https://ingewortel.github.io/2022-Tcell-evolution/ with an interactive simulation). While this environment is not representative of what T cells encounter *in vivo*, it allowed us to see which migratory patterns could

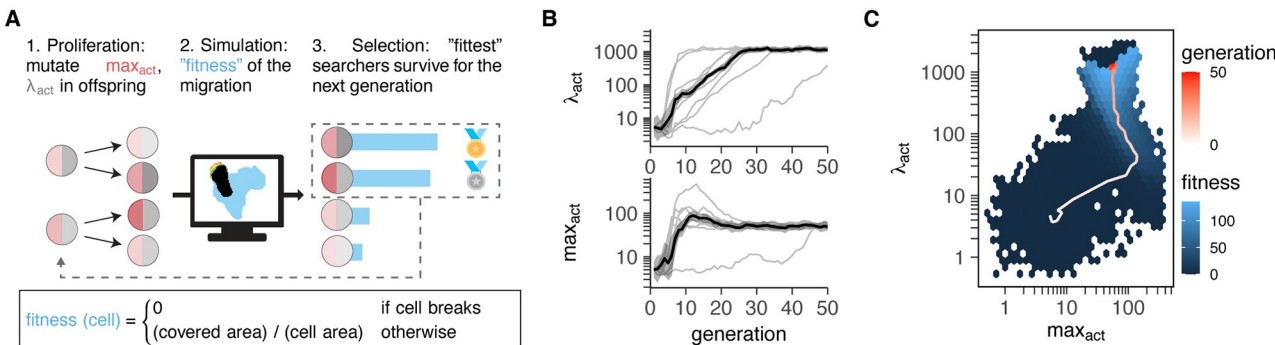

**Fig 2. Evolution of optimal search behaviour in Act cells is subject to constraints and trade-offs.** A: Simulated evolution in an evolutionary algorithm. A population of 10 Act cells with their own ($max_{act}$,$\lambda_{act}$) parameters each produce three daughter cells with randomly "mutated" parameters (see Methods for details). After simulating migration for all 40 cells, only the 10 "fittest" cells (i.e. those that explored the largest area) survive as the next generation. Here, fitness is defined as the total area explored by the simulated cell during the simulation (normalized by the area of the cell itself). The fitness is zero if the cell breaks during the simulation. B: Evolution of $\lambda_{act}$ and $max_{act}$ over 50 generations. Black line + shaded area shows the mean ± SD within each generation. Thin gray lines show the same curve for 9 other, independent runs. C: Evolution of $\lambda_{act}$ and $max_{act}$ in the context of the median "fitness" experienced by cells with those parameters. The red trajectory represents one single run; the (blue) fitness landscape is constructed by averaging measured fitnesses from all cells of all (10) independent runs at given parameters.

evolve without any constraints from the tissue. Since the evolutionary objective (having a large fitness) requires cells to explore a large area (Fig 2A), the theoretical optimum in this scenario is simply to maximise both speed and persistence—so as to move as far as possible without turning and visiting the same area twice [37].

To see if this theoretical optimum could arise through evolution of $max_{act}$ and $\lambda_{act}$, we started with a population where both parameters were too low for active cell migration (S1(A) Fig) and allowed them to evolve. During the evolutionary run, the average values of $max_{act}$ and $\lambda_{act}$ in the population gradually increased before eventually plateauing at values of 50 and 1165, respectively (Fig 2B). Strikingly, the same stable endpoint was reached in 9 out of 10 independent runs (Fig 2B; the last run did not fully converge but nevertheless moved towards that same point). This end point was associated with the highest fitness (Fig 2C)—suggesting that this parameter combination was somehow optimal. These results demonstrate that some form of evolutionary adaptation is taking place.

## Constraints and trade-offs alter evolved Act-cell search patterns

To investigate how the evolved migratory behaviour arose, we next analysed motion at different parameter combinations along the evolutionary trajectory and surrounding the evolved optimum (Fig 3). The increase in the motility parameters $max_{act}$ and $\lambda_{act}$ coincided with an increase in migratory ability over the generations as measured by the average explored area (Fig 2C) as well as speed and persistence (Fig 3A). Along the trajectory, motion was well-described by a persistent random walk (S2 Fig).

Yet, intriguingly, cells at the empirical optimum did not have the theoretical maximum speed and persistence (Fig 3B). Whereas cells could still reach higher speeds by further increasing $\lambda_{act}$, the higher force on the cell membrane also caused frequent cell breaking (Fig 3B). Thus, the "optimum" in the fitness landscape that cells converge to reflects the point where cells achieve maximum speed while still conserving their integrity (S1(B) Fig). Likewise, there appears to be a trade-off between speed and persistence at parameters surrounding the optimum: the increased speeds observed at higher $\lambda_{act}$ and lower $max_{act}$ values come at the cost of a lower persistence (Fig 3B). This conflict likely arises because the evolved "optimal" cell is already quite broad and persistent (S1 Movie). We have previously shown that both speed and persistence saturate as the cell broadens [34]. In this "saturation regime" of the UCSP, an even higher persistence requires a large effort to maintain a stable, broad protrusion, slowing the

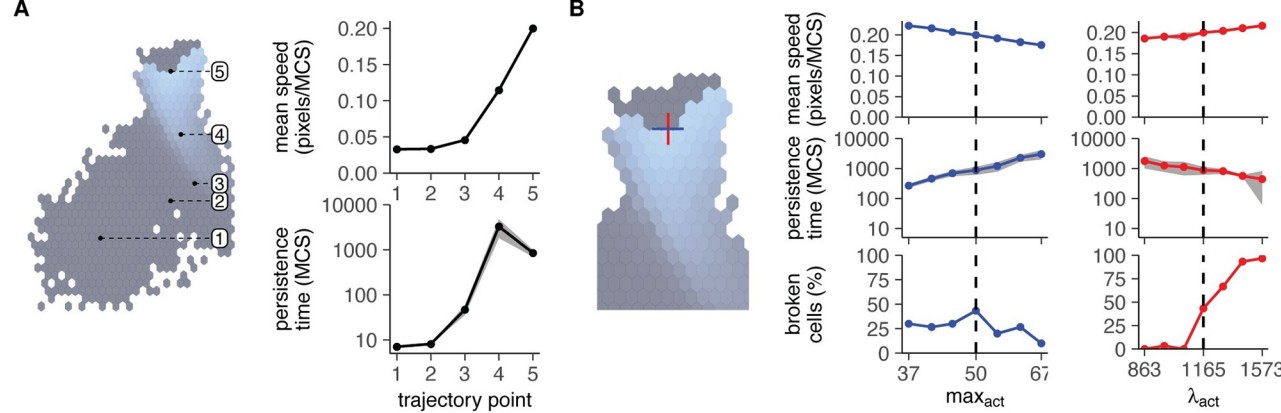

**Fig 3. Evolution of optimal search in Act cells is subject to constraints and trade-offs.** A: Speed and persistence measured at different points in the fitness landscape of Fig 2C. B: Speed, persistence, and cell breaking measured around the evolved optimum ($max_{act}$ = 50, $\lambda_{act}$ = 1165).

cell down [34]. Because of this trade-off, the cell evolves towards parameters where it is persistent enough that it rarely visits the same area twice (S1 Movie), yet not so persistent that this comes at the cost of a low speed (Fig 3A and 3B).

All in all, as expected, cells evolved their migration parameters to cover larger areas by increasing both their speed and persistence. However, the point that they eventually converged to was altered by constraints—in this case the inability to further increase speed without breaking (S1 Fig), and the inability to further increase persistence without decreasing speed (Fig 3B).

To further illustrate the importance of these constraints, we also simulated evolution in a model where the mapping from model parameters to speed and persistence is much more direct [4] (see Methods). In this model, cells are circles that alternate between pauses (of duration $t_{pause}$) and intervals of straight motion (with duration $t_{free}$ and speed $v_{free}$, directly reflecting persistence time and speed). In this model, we found that as expected, cells simply increased both their speed and persistence time to increase their fitness (S3 Fig).

These results show that—especially when genotype-phenotype mapping gives rise to trade-offs—the evolved migratory traits (such as speed and persistence) may be only partially determined by their theoretical optima.

## Environmental constraints, not evolved cell-intrinsic parameters, are the major determinants of Act-cell migration patterns in tissues

Finally, we examined how environmental constraints affected the search behaviours evolved by cells. We therefore repeated the evolution experiment, but now assessed fitness by simulating T-cell migration inside a tissue instead of empty space (Fig 4A).

We focus on the epidermal layer of the skin, where skin-resident T cells continuously patrol to search for signs of re-infection by foreign invaders [20]. Because of the skin's barrier function, the keratinocytes in the epidermis are very tightly packed, forming an extreme example of a restrictive environment. We have previously shown that such restrictions strongly affect cell motion, for example by obscuring the UCSP [34].

Because migration in a stiff tissue requires higher $\lambda_{act}$ forces [34], we started with a population with slightly higher $\lambda_{act}$ than before, which was still low enough to prevent active

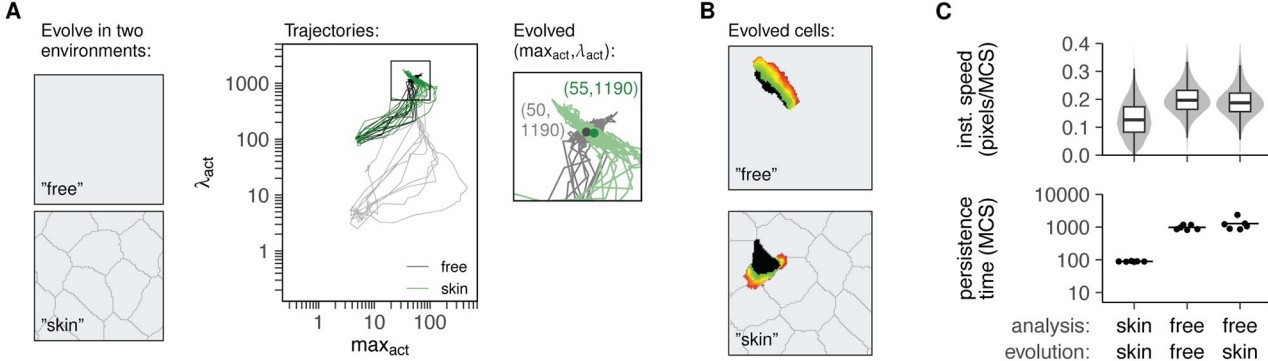

**Fig 4. Act cells in different environments evolve similar parameters but different shapes and behaviour.** A: Evolution trajectories of the ($max_{act}$, $\lambda_{act}$) parameters compared between different runs of evolution in empty space ("free", see also Fig 2) and evolution in a rigid simulated tissue ("skin"). Black lines represent the "free" cells evolved from the higher $\lambda_{act}$ = 100, while gray lines show the trajectories from Fig 2. Zoomed square shows where parameters converge in the two different environments after 50 generations, at similar ($max_{act}$,$\lambda_{act}$) values. B: Cells evolved in different environments have similar parameters but different shapes and behaviours. See also S1 Movie. C: Speed and persistence of cells with parameters evolved in simulated skin ("skin-skin"), parameters evolved in an empty environment("free-free"), or parameters evolved in simulated skin but analysed in an empty environment ("free-skin"). Speeds are represented as instantaneous speeds of each individual step in the simulation, and persistence times reflect 6 independent measurements at the same parameters (see Methods for details).

migration (S4(A) Fig). $\lambda_{act}$ and $max_{act}$ values once again increased during evolution, and this was associated with an increase in the fitness and the area explored (Figs 4A, 1B and 1C). All runs once again converged to roughly the same endpoint—this time with $max_{act}$ = 55, $\lambda_{act}$ = 1190, only slightly different from the endpoint reached by evolving the same cells in a free environment ($max_{act}$ = 50, $\lambda_{act}$ = 1165, Fig 4A). The small difference was not a result of the higher $\lambda_{act}$ starting value, because when we used the same starting value for evolution in empty space, cells still stabilised at very similar parameters ($max_{act}$ = 50, $\lambda_{act}$ = 1190, Figs 4A and 1B).

All in all, the parameters that cells evolve depends on the fitness landscape they experience. This landscape differed slightly between cells evolved in the free environment and cells evolved in the simulated skin, where the presence of surrounding tissue affects their ability to explore area and stay intact (S4(C) Fig). Still, the end result was more or less the same: cells evolved towards very similar parameters and once again could *not* reach maximum speed because of increased cell breaking at high $\lambda_{act}$ values (S4(D) Fig). Rather, they again evolved to a level of speed and persistence that allowed them to keep moving within a rigid environment without breaking apart (S1 Movie).

Because cells evolved towards remarkably similar parameters regardless of their environment (Fig 4A), we next asked to what extent these small parameter differences still altered their behaviour. Indeed, these small parameter differences still corresponded to very different shape and behaviour (Fig 4B and S1 Movie). Cells evolved in the skin had both a lower speed and persistence than "free" cells in empty space (Fig 4C). However, these altered migration statistics are likely due to the altered environment, not just the slight differences in cell-intrinsic parameters. To confirm this, we took cells with parameters optimised for migration in the skin ($max_{act}$ = 55, $\lambda_{act}$ = 1190) and analysed their motility in an empty environment. These cells were much more similar to the "free" evolved cells—despite having been optimised in a different environment (Fig 4C and S1 Movie). Thus, even if cells can optimise their search efficiency to some extent by evolving cell-intrinsic motility parameters, their eventual migration statistics may still be shaped largely by environmental constraints.

## Discussion

### T-cell search and the debate on optimal foraging

The interpretation of the diversity in T-cell motility patterns as optimised search strategies is remarkably similar to the so-called optimal foraging theory, used by evolutionary biologists to study the ways in which animals search for food [13, 21]. The general idea is that animals (or immune cells) adopt migratory patterns that maximise their ability to find food (or antigens) [22, 25, 37, 38]. Given the implicit assumption that the behaviour we see has somehow been selected during evolution, we seek specific functions the animal (or cell) could have optimised its behaviour for.

Yet even if a certain migration characteristic is optimal or beneficial in some context, this does not prove that it has truly evolved to aid immune system function [39]. It might also have arisen as a side effect of some other process [40], or simply because other migration modes are impossible within the relevant constraints. Ignoring this possibility may lead to spurious interpretations and hamper a true understanding of immune cell migration—especially since it is almost always possible to come up with a context in which the observed pattern would indeed be beneficial [13]. Applying optimality reasoning can thus be misleading when the pattern in question was only partially the result of evolutionary optimisation.

We therefore used a CPM to redefine the baseline expectations for T-cell migration behaviour given a realistic intracellular migration machinery, cell shape, and environment. The dynamic interactions in this model yield an indirect genotype-phenotype mapping that

introduces non-trivial trade-offs and constraints in the speeds and persistences that cells can obtain [34]. This allowed us to use this model to simulate an evolutionary process where we let cells maximise a very simple objective function: cover as much area as possible. This problem is analogous to destructive search, for which the theoretical optimum behaviour is simply to move as fast and as straight as possible [37]. Nevertheless, cells did not reach the theoretical optimum because of the constraints and trade-offs naturally arising in the CPM. This problem is exacerbated when cells are placed in a tissue where surrounding cells pose the dominant constraints on cell motility: cells with very similar parameters, optimised for the exact same function, move in completely different patterns depending on their environment. These results once again highlight that complex migration features can emerge spontaneously from environmental constraints and are not necessarily "strategies" adapted for some specific function.

## Model limitations

In our simulations, we used a simplified model of cell migration in which actin dynamics are described implicitly through a positive feedback on migration. We found that trade-offs arose that impacted the migration patterns cells could evolve. But the question remains: to what extent can these results be generalised to living cells?

Although the Act-CPM is a simple model, it nevertheless captures several of the opposing forces expected to govern cell behavior. Like many other models of cell migration, it combines the concepts of "local activation" (i.e., the positive feedback on protrusion activity) and "global inhibition" (i.e., the opposing force from membrane tension) to model cell motion. This general mechanism appears sufficient to capture shape-motility relations similar to those in other model systems (see the discussion in [34]). Indeed, the trade-off between speed and persistence at the evolved optimum arises because both speed and persistence saturate around these parameter values, and such saturations have been observed in other models as well [41, 42]. Likewise, the trade-off between higher protrusion forces (leading to higher speeds) and more frequent cell breaking is rarely modelled explicitly—yet it seems plausible that there is a limit to the forces cells can exert on their membrane, and cell breaking has indeed been observed for leukocytes migrating at high temperatures [43], as well as NK cells and T cells deficient for DOCK8 [44]. Thus, it seems likely that at least some of the constraints and trade-offs arising in the Act-CPM might also apply to real cells.

Nevertheless, the Act-CPM remains a simplification, and real cells might be subject to other constraints that are not captured in this model. An interesting direction of future work would be to compare evolved search patterns in different models, especially those in which cell shape and motility are emergent rather than imposed by the user [45–51].

Aside from the migration model, we used an evolutionary setting that was in itself highly simplified. For example, our evolutionary algorithm ignores that T cells may have to optimise different functions simultaneously, might suffer from exploration-exploitation trade-offs [5], and do not evolve individually but as part of a larger organism. Yet even in this very simple setting, we find that evolving towards the theoretical optima for speed and persistence is not a trivial task, resulting in migration patterns that are part optimised "strategy" and part "side effect" from the mechanistic constraints experienced by the cell.

## Adaptation, (co-)evolution, and the environment

When letting cells evolve in different environments, we found that observed migration patterns were strongly affected by environmental constraints. But our simulated cells only had to optimise their behavior for one environment at a time—while a key feature of T cells is their ability to migrate in almost any environment.

Indeed, T cells are known to adapt their migratory behavior to both tissue and context [52]. For example, they migrate vigorously in the paracortex of the lymph node [1], but can be forced to move much more slowly in stiff peripheral tissues [20]; and chemokinesis of memory T cells in response to CCL21 was shown to be faster than that of naïve T cells [53]. In fact, this adaptability may itself have been the key feature that T cells have evolved. This would essentially make the evolution of T-cell search a multi-objective optimisation problem, adding yet another layer of complexity to the process of evolving search strategies (the interplay between evolution and plasticity has been studied extensively in the context of adaptation to new environments; see, for example, [54–56]). Since unexpected migration patterns already emerge even in our highly simplified simulations, the addition of adaptation would only make it more important to ask which features of migration have been optimised and which reflect a compromise.

Another option is that the environment itself may have co-evolved with T cells to optimise T-cell search. This might be true especially for lymphoid organs—indeed, the fibroblastic reticular cell (FRC) network in lymph nodes has been extensively studied [57], and has been suggested to facilitate the search for antigen by naïve T cells [3].

Yet such co-evolution of tissues to facilitate T-cell migration is debatable in non-lymphoid organs (whose primary function is unrelated to T-cell function). For example, it seems unlikely that the brain has evolved a structure specifically to facilitate Lévy flights of T cells, even if these are beneficial when searching for rare pathogens [22]. And while a less rigid structure of the skin epidermis studied here would likely make it easier for tissue-resident T cells to patrol for invaders [34], such a tissue is unlikely to evolve because it would also make it easier for said invaders to enter in the first place. Thus, even with the option of environmental co-evolution, it remains important to consider environmental context when evaluating search strategies.

## Evolved "strategies" are not always optimal

To our knowledge, this is the first study applying the "bottom-up" approach to evolution in the context of T-cell search. Importantly, we observed migratory behaviours that have previously been attributed to optimality for some T-cell function but are clearly not optimal in our artificial evolution setting. For example, the existence of intervals of fast and slower motion in T-cell tracks has been attributed to an "intermittent search strategy" where T cells balance area exploration (through fast movement) with local exploitation (slower movement and more frequent turning) [5, 6, 25]. Yet we see that variations in speed occur naturally in Act cells: protrusion dynamics automatically yield intervals of slower and faster (or even stop-and-go) motion. In simulated skin, cells move fast when they are moving forward between two keratinocytes, but then must slow down temporarily when they reach a junction and have to choose a direction. Thus, even though cells in our simple evolutionary experiment gain fitness only from exploring area—and not from exploiting it—their motion nevertheless resembles an intermittent search strategy. This observation mirrors an earlier finding that the "stop-and-go" behavior of T cells in lymph nodes might arise through interactions with the environment rather than an earlier suggested cell-intrinsic clock [19]. Thus, intermittent search "strategies" can also arise naturally through cell-intrinsic migration dynamics or through the environment, even in cases where they do not benefit any specific function at all. This highlights how simulating the process of evolution explicitly can provide important context when interpreting T-cell search patterns in the light of optimality.

## Future directions

While the application of optimality theory to animal foraging has provoked considerable criticism [13], the same line of reasoning is applied with far less debate in the context of T-cell

migration [5, 58]. Yet our results strongly suggest that the criticisms against optimal foraging theory are also relevant for the interpretation of T-cell search patterns. Do T cells in the brain display Lévy-like statistics because that helps them catch rare pathogens [22], or because they are forced to do so by a combination of their cell-intrinsic migration machinery and the structures they are navigating in the brain? Would they adopt a different pattern in the same environment when fighting a more prevalent pathogen, or would they maintain the same migration mode even when it is no longer beneficial? We therefore suggest using models like the CPM [33, 34, 59, 60], where migration patterns arise naturally from an interaction between the cell and its environment rather than being imposed, to define the baseline expectations for T-cell search in a "bottom-up" approach. By investigating which migratory characteristics emerge without being optimal or even beneficial, we can zoom in on the motility aspects that have truly been evolved to assist immune system function—without being misled by features that are merely inevitable side effects of an intracellular machinery acting in a complex environment.

## Methods

### Act-CPM

For our simulations, we used the Act-CPM [33]. For more information, we refer to the relevant literature [33, 34], but a brief description follows below.

The Act-CPM extends the CPM as follows. Every monte carlo step (MCS, the time unit of the CPM), pixels try to "steal" pixels away from neighbouring cells by copying their identity into that pixel. The success probability $P_{copy}$ of these copy attempts depend on the *Hamiltonian* (global energy), which consists of different terms:

$$H = H_{adhesion} + H_{volume} + H_{perimeter} + H_{act} \tag{1}$$

Here, $H_{adhesion}$ assigns an energetic penalty to each pair of neighbouring pixels $(i,j)$ on the grid that do not belong to the same cell. Likewise, $H_{volume}$ and $H_{perimeter}$ control the cell's size and circumference via a penalty that depends quadratically on the deviation from some "target" value $X_i^*$:

$$H_X = \sum_{i \in cells} \lambda_X (X_i - X_i^*)^2 \tag{2}$$

In practice, we mostly look at the energy difference $\Delta H$ a candidate copy attempt would introduce, rather than considering the absolute energy $H$.

The Act-CPM extends $\Delta H$ with a positive feedback term, such that pixels newly gained by the cell retain an elevated "protrusive activity" for a period of $max_{act}$ MCS. This is reflected by the negative (=energetically favourable) $\Delta H_{act}$ assigned to copy attempts that go from a more active source pixel $s$ into a less active target pixel $t$:

$$\Delta H_{act}(s \rightarrow t) = -\frac{\lambda_{act}}{max_{act}}(GM_{act}(s) - GM_{act}(t)) \tag{3}$$

Here, $GM_{act}(p)$ represents the geometric mean of the activity values in the (Moore) neighbourhood of pixels $p$. $\Delta H_{act}$ is negative when $GM_{act}(s) > GM_{act}(t)$. Details on parameters used will follow below.

## Simulations

All CPM simulations were performed with Artistoo [61]. Simulations were performed for 10,000 MCS.

**Initialisation.** For simulations of "free" T cells moving in an open space, cells were seeded in the middle of a 150x150 pixel grid with periodic boundaries, and allowed a burnin time of 500 MCS to gain their optimal volume and shape.

For simulations of T cells moving in the epidermis, 31 keratinocytes were seeded randomly on a 150x150 pixel grid with periodic boundaries. To ensure proper formation of the tightly packed keratinocyte layer, cells were initially seeded with a tighter perimeter of 200 (making them rounder and preventing cell breaking). Each cell was allowed to grow for 50 MCS before the next cell was seeded, also to ensure that cells did not become entangled and break. After seeding all keratinocytes, the tissue was given 500 more MCS to equilibrate, after which the first keratinocyte was replaced by a T cell and the keratinocytes were given their true perimeter value (see section *CPM parameters* below).

**CPM parameters.** Parameters were selected from [34], allowing realistic shapes and migration behaviour without the cells falling apart (Table 1). Only $max_{act}$ and $\lambda_{act}$ were varied during the evolutionary runs and in the simulations analysing speed and persistence (see below); other parameters were held constant.

**Act-CPM parameters.** In the simulations of evolution, $\lambda_{act}$ and $max_{act}$ were not specified, but evolved spontaneously during the evolutionary run (see section *Evolution of optimal migration modes* below).

To assess speed and persistence at points of interest in the fitness landscapes, simulations were performed at fixed combinations of $max_{act}$ and $\lambda_{act}$. Along the evolutionary trajectory, simulations were performed at $(max_{act}, \lambda_{act}) = (5,5), (50,17), (110,30), (70,150)$, and $(50,1185)$. To examine the behaviour around the optimum, simulations were performed at points surrounding the optimum $(max^{*}_{act}, \lambda^{*}_{act})$ as:

$$\{(max^{i}_{act}, \lambda^{*}_{act}), (max^{*}_{act}, \lambda^{i}_{act})\} \tag{4}$$

with:

$$max^{i}_{act} = \exp[\log(max^{*}_{act}) \pm i \cdot 0.1], \qquad \lambda^{i}_{act} = \exp[\log(\lambda^{*}_{act}) \pm i \cdot 0.1], \qquad i \in \{0, 1, 2, 3\} \tag{5}$$

The "optimum" $(max^{*}_{act}, \lambda^{*}_{act})$ from multiple evolutionary runs was determined as follows. First, runs were removed if they had not "converged" to any optimum (i.e., if their population average of both parameter values changed by $\geq 20\%$ over the last 10 generations). Parameter

**Table 1. CPM parameters used in the "free" and "skin" environments.**

| Parameter | Free simulations | Skin simulations |
|---|---|---|
| Temperature | 20 | 20 |
| Volume (pixels) | 500 | 500 (750) |
| $\lambda_{Volume}$ | 30 | 30 (30) |
| Perimeter | 260 | 260 (330) |
| $\lambda_{Perimeter}$ | 2 | 2 (10) |
| Adhesion cell-background | 20 | 20 (20) |
| Adhesion keratinocyte-keratinocyte | - | 200 |
| Adhesion T cell-keratinocyte | - | 2 |

Skin simulation parameters refer to the T cells; keratinocyte parameters are inside the brackets.

values of each remaining run were then averaged over the last 10 generations (focusing only on the 10 fittest individuals and pooling values from different runs) and rounded to the nearest 5. Because populations evolve their parameters on a logarithmic scale (see Eq 6 below), this averaging was performed on a log scale as well.

## Evolution

To simulate evolution of optimal migration parameters $\max_{act}$ and $\lambda_{act}$, we used a genetic algorithm as described below. Ten independent runs were performed in every experiment. Simulations in skin were performed in the "stiff" tissue from [34].

**Evolution of optimal migration modes.** To simulate evolution, we started with a population of $N_{pop} = 10$ Act cells. For simulations of cells in an empty environment, cells in the initial population had $\max_{act} = \lambda_{act} = 5$ (for simulations of cells in the epidermis, initial T cells had the same $\max_{act} = 5$ but a higher $\lambda_{act} = 100$ because of resistance from the surrounding tissue). The following steps were then repeated for a total of 50 generations:

1. Population growth: $\lambda = 3$ offspring cells were generated from each of the $N_{pop}$ cells in the population, with mutated $\max_{act}$ and $\lambda_{act}$ parameters (see section *Mutation* below).

2. Simulation of migration: Each of the $(\lambda + 1)N_{pop}$ cells in the resulting population was simulated independently for 10,000 MCS as described previously, yielding a "fitness" for each cell (see section *Fitness* below).

3. Survival of the fittest: Individuals in the population were ranked according to fitness, and only the $N_{pop}$ fittest individuals survived for the next generation.

These choices of $N_{pop}$ and $\lambda$ are somewhat arbitrary; they do not affect our qualitative conclusions on what happens during evolution, but they may affect how long the evolutionary process takes to converge. For example, a larger population allows faster and more thorough exploration of the parameter space. The $\lambda$-dependent selection strength determines how long "reasonably fit" individuals can remain in the population and thus have the chance to further evolve. For computational efficiency, we here chose values that allowed evolution to occur within a reasonable number of iterations.

**Mutation.** For mutation of $\max_{act}$ and $\lambda_{act}$ parameters of a given cell, parameter values $x$ were first log-transformed and subsequently mutated with a random error term:

$$x_{mut} = \ln x + \epsilon \tag{6}$$

where

$$\epsilon \sim \mathcal{N}(\mu = 0, \sigma_{mut}) \tag{7}$$

using $\sigma_{mut} = 0.6$ for the first 5 simulations and $\sigma_{mut} = 0.2$ afterwards. The higher initial choice of $\sigma_{mut}$ is for efficiency reasons only; note that at the initial low values of $\lambda_{act}$ and $\max_{act}$, cells do not actively move and slight changes in these parameter values can therefore not affect fitness. Only through genetic drift do cells escape this "fitness plateau" into the motile regime where $\lambda_{act}$ and $\max_{act}$ are high enough for migration. Choosing a higher initial $\sigma_{mut}$ speeds up this process. It does not affect the main results or conclusions from the simulation but simply reduces the number of generations it takes for cells to escape the fitness plateau and start evolving.

**Fitness.** Cells were given a fitness of 0 if they "broke" (connectedness <90% at any point in the 10,000 MCS simulation, see section *Cell breaking*). Otherwise, their fitness equaled the area covered during the simulation (measured in the number of cell volumes of 500 pixels).

### Analysis

In the simulations used to compute speed and persistence, the position of the cell's centroid was logged every 5 MCS to produce cell tracks. The cell's integrity was also measured to ensure that cells stayed intact.

Simulated tracks were then analysed in R (version 3.4.4) using the celltrackR package (version 0.3.1) [62]. Speed and persistence were computed in a step-based analysis on 6 groups of 5 simulated tracks (see below), which yielded 6 independent estimates for every parameter combination, from which the mean and SD were assessed. For analysis of mean squared displacements and autocovariance, see below.

**Speed.**   To compute speeds, we first computed instantaneous, "step-based" speeds along cell tracks (using the "speed" function of celltrackR). The average of this distribution was then reported as the mean speed.

**Persistence.**   The persistence time of moving cells was computed from the decay in the autocovariance curve as described previously in [34].

**Cell breaking.**   To quantify cell breaking at a given $\max_{act}$ and $\lambda_{act}$ combination, we counted the percentage of simulations in which the minimum *connectedness* (C) was <90%, as described previously [34].

For details, we refer to [34], but briefly: C measures the probability that two random pixels of a cell are part of a single, unbroken unit. This measure ensures that an intact cell (which has only $n = 1$ connected component) gets $C_i = 1$, whereas a cell broken in many parts ($n > > 1$) gets a very low connectedness. It also means that a single pixel breaking off a cell does not have a huge impact on connectedness, whereas a cell splitting in two equal parts does (even though $n = 2$ in both cases).

**Mean squared displacement (MSD) curves.**   Mean squared displacement plots were computed in celltrackR [62] (there are multiple, subtly different methods to compute MSD curves; we used: *aggregate(tracks, squareDisplacement)*). To compare these curves to the persistent random walk model, Fürth's equation [63, 64] was fitted to these data:

$$\mathrm{MSD}(\Delta t) = 4D \cdot (\Delta t - P(1 - e^{-\Delta t/P})) \tag{8}$$

where $\Delta t$ is the time interval over which displacements are considered, and the persistence $P$ and diffusion coefficient $D$ are the parameters to be fitted.

To fit these curves robustly, some technical points must be considered. First, there are many more ways to extract small intervals $\Delta t$ from any given track than there are to extract long ones. In general, if a track contains $n$ steps between $t = 0$ and $t = n$, there are $n$ displacements for $\Delta t = 1$ and just one for $\Delta t = n$. Also note that for larger $\Delta t$, many of these overlap: for instance, for $\Delta t = n - 1$ we have two displacements ($t = 0 \rightarrow n - 1$ and $t = 1 \rightarrow n$), but these overlap almost entirely since they both contain the data between $t = 1$ and $t = n - 1$. Thus, they are not independent observations. As $\Delta t$ increases, we have fewer independent observations of the MSD and thus larger uncertainty in the data. To obtain robust fits, we therefore weighted each data point ($\Delta t$, MSD($\Delta t$)) by the number of *independent* displacements the MSD was based on.

Second, CPM cells move on a discrete grid. At timescales far below their persistence time, they only move stochastically—but given the discrete nature of the grid, these very small displacements deviate from the (continuous) Fürth equation and give artefacts when fitting MSD curves. When we are fitting the MSD curve, we are mostly interested in the behaviour around and beyond the persistence time $P$. We therefore fitted curves in two steps:

1. First, a very rough fit was performed on the data. The fitted parameters $D_0$ and $P_0$ are not accurate for the reasons mentioned above, but they *are* at least in the right order of

magnitude. The estimate $P_0$ was then used to discard data points with $\Delta t < P_0$; thus, we fit only the data at timescales where the cell is actually moving.

2. All points with $\Delta t \geq P_0$ were then used for the final fit using the R function *nls*, setting *"weights"* as described above. Since we are interested in scaling behaviour here and typically consider the MSD on a logarithmic scale, we also perform the fitting on a logarithmic scale: *log(MSD) ∼ log(4\*exp(logD)) + log((dt—P\*(1- exp(- dt/P))))* To help the algorithm converge, we fit the logarithm of $D$ rather than $D$ itself, providing the estimated log $D_0$ and $P_0$ to the algorithm as starting point.

**Autocovariance curves.** Autocovariance curves were computed in celltrackR [62] (using: *"aggregate(tracks, overallDot)"*). For persistent random walks, autocovariances should decay exponentially with time interval $\Delta t$:

$$\text{autocovariance}(\Delta t) = c \cdot e^{-\Delta t/P} \tag{9}$$

with $c$ a constant, and $P$ the persistence time.

Once again, we run into the problem that data at very small $\Delta t$ can cause problems because CPM cells, at that scale, do not actually move (see explanation for MSD curves above). We therefore focused on $\Delta t$ values that were not too small, filtering $\Delta t > 0.5 P_{\text{MSD}}$ (with $P_{\text{MSD}}$ the persistence estimate from the MSD fit).

Given the duration of our simulations, data span a large range of $\Delta t$ values; however, at low persistences, the autocovariance rapidly decays to zero. If we were to include all the data up to very large $\Delta t$, most of these data points would then just contain noise around an autocovariance of $\sim 0$ and this noise would dominate the fit. To circumvent this problem, we considered the point $t_{5\%}$, which is the smallest $\Delta t$ for which the autocovariance drops below 5% of its initial value. We then filtered points for which $\Delta t < 3 t_{5\%}$. (The exact choice of this threshold is somewhat arbitrary and does not really matter; the point is that we are looking at a range of $\Delta t$ values where the autocovariance is actually decaying).

Finally, we fitted the exponential decay equation using R's *nls* and formula: *autocovariance ∼ c \* exp(—dt / sqrt(Psq))* where we fit Psq = $P^2$ rather than $P$ itself to prevent the algorithm from considering negative $P$ during the fitting procedure, and provide ($c = 1$, Psq = $P_{\text{MSD}}^2$) as a starting point to help the algorithm converge.

## Evolution in the Beauchemin model

**Model.** The Beauchemin model describes T-cell migration in the lymph node [4]. Cells follow a variation of a random walk where they alternate between "runs" of duration $t_{\text{free}}$ (where they move at speed $v_{\text{free}}$), and pauses of duration $t_{\text{pause}}$ (where they do not move and can change direction).

**Simulations.** Simulations were performed in celltrackR [62] using the function "beaucheminTrack", with a simulation time of 10000 steps and time resolution $\Delta t = 1$ step to mimic CPM simulations. The parameters $v_{\text{free}}$, $t_{\text{free}}$, and $t_{\text{pause}}$ were each allowed to evolve (see below). All other parameters were kept default.

**Evolution.** The parameters $v_{\text{free}}$, $t_{\text{free}}$, and $t_{\text{pause}}$ were allowed to evolve from an initial value of $v_{\text{free}} = t_{\text{free}} = t_{\text{pause}} = 1$, using the same dynamics as with evolution of the Act-CPM (using the same $\lambda = 3$ and $N_{\text{pop}} = 10$ as before, but now only for 25 generations, after which fitness did not further increase). To mimic evolution of the Act-CPM, $\sigma_{\text{mut}}$ was again initialised at 0.6 and reduced to 0.2 after the first 5 generations (although we note that this choice is not

very important for this model, because changes to the model parameters immediately translate to fitness changes in this case).

**Fitness.** For convenience, we measure fitness as area explored in a discretised space. At every time point $\Delta t$, we count the number of (imaginary) pixels that would fall within a radius $r$ of the cell's coordinate (only counting pixels towards the fitness if they have not been explored before). For comparison with the Act-CPM results, we use a radius $r = 13$ (giving a cell area of $\pi r^2 \approx 530$ pixels, approximately equal to cell area in the Act-CPM), and divide the area explored by this area to obtain the fitness (since cells cannot break in this model, the fitness is now directly proportional to area explored).

## Supporting information

**S1 Movie. Tissue constraints impose different migration patterns in evolved cells with very similar parameters.** Cells migrating in a free environment are broader and move more persistently. This movie is available online at: https://ingewortel.github.io/2022-Tcell-evolution/. (MP4)

**S1 Fig. Fitness landscape experienced by cells evolving their $max_{act}$ and $\lambda_{act}$ values.** A: Act cells with $max_{act} = 5$ and $\lambda_{act} = 5$ cannot actively move. Distrubutions of instantaneous speeds equal those of control cells with $\lambda_{act} = 0$ (which cannot form protrusions by definition). B: Fitness landscape plots show mean fitness (area explored measured in the number of cell target areas of 500 pixels; broken cells have a fitness of zero), mean area searched by non-broken cells, and percentage of broken cells for different ($max_{act}, \lambda_{act}$) combinations. Gray fields represent a value of zero. (TIF)

**S2 Fig. Motion statistics for cells along the evolutionary trajectory.** Motion was analysed for the points (2–5) along the evolutionary trajectory of Fig 3 (point 1 was skipped since at these parameters, cells do not yet move). A: Mean square displacement (MSD) curves of simulated tracks (solid) and the persistent random walk (P-RW) fit (dashed), for points 2–5 along the evolutionary trajectory. B: Autocovariance curves of the simulated tracks (mean ± interquartile range, gray) and an exponential decay fit (red, autocovariance $\sim \exp{-\Delta t/\tau}$)). The dashed vertical lines represent the corresponding fitted value of $\tau$, which is another measure of persistence time. (TIF)

**S3 Fig. Evolution without constraints in the Beauchemin model.** To illustrate the importance of a non-trivial mapping from model parameters to speed and persistence, we simulated evolution in the Beauchemin model of T-cell migration in the lymph node [4]. In this model, cells alternate between "runs" of duration $t_{free}$, during which cells move at speed $v_{free}$, and pauses of duration $t_{pause}$, where cells do not move and can change direction. Thus, there is a more or less direct mapping from model parameters to migratory behavior, with $v_{free}$ directly determining cell speed and $t_{free}$ acting as a persistence time. At each step of fixed duration $\Delta t$, we check how much new explored area is within radius $r = 13$ of the cell's current location. A: Fitness over generations (10 independent runs). After about 15 generations, the fitness stops increasing. This is because the fitness function measures area covered within radius $r$ at every (discrete) time point where the cell is measured (determined by the time resolution $\Delta t$). Once the cell is fast and persistent enough that two subsequent circles do not overlap, and that it never returns to the same circle, it reaches a maximum fitness. This corresponds to a fitness plateau where cells can perform a random walk in parameter space. B, C: Evolution of the three independent model parameters over time. Independent runs follow the same trend

(predictably increasing $t_{\text{free}}$ and $v_{\text{free}}$, while decreasing $t_{\text{pause}}$ which does not help them explore area). But unlike with the Act-CPM, they do not converge to exactly the same end state (likely due to the fitness plateau observed in panel A). In panel C, trajectories are color-coded from dark (early generations) to light blue (later generations). D: Trajectory of one example run shown in context of the fitness landscapes for each possible pair of two parameters. This again shows that parameters first evolve towards fast, persistent motion. Once a good fitness is reached, they follow a random walk in the parameter space. Analogous to earlier figures, the "fitness" equals area explored divided by circle area ($\pi r^2 \approx 530$ pixels), making the fitness directly proportional to area explored.
(TIF)

**S4 Fig. Fitness landscape experienced by cells evolving in a rigid "skin" environment.** A: Cells with $\max_{\text{act}} = 5$ and $\lambda_{\text{act}} = 5$ or $\lambda_{\text{act}} = 100$ cannot actively move in the rigid skin tissue. Distrubutions of instantaneous speeds equal those of control cells with $\lambda_{\text{act}} = 0$ (which cannot form protrusions by definition). B: Fitness landscape showing median fitness and example trajectories for cells evolved in an empty environment ("free", two trajectories are shown with a different starting point) compared to cells evolved in stiff tissue ("skin"). C: Fitness landscape showing mean fitness, mean area searched, and percentage of broken cells (see also S1 Fig). D: Mean speed, persistence, and cell breaking of Act cells in simulated skin at parameters surrounding the evolved optimum ($\max_{\text{act}} = 55$, $\lambda_{\text{act}} = 1190$). The square represents a zoomed version of Fig 4A showing this optimum.
(TIF)

## Acknowledgments

The authors thank Nir Gov for useful discussions.

## Author Contributions

**Conceptualization:** Inge M. N. Wortel, Johannes Textor.

**Data curation:** Inge M. N. Wortel.

**Formal analysis:** Inge M. N. Wortel.

**Funding acquisition:** Johannes Textor.

**Investigation:** Inge M. N. Wortel.

**Methodology:** Inge M. N. Wortel, Johannes Textor.

**Project administration:** Johannes Textor.

**Resources:** Johannes Textor.

**Software:** Inge M. N. Wortel, Johannes Textor.

**Supervision:** Johannes Textor.

**Validation:** Inge M. N. Wortel.

**Visualization:** Inge M. N. Wortel.

**Writing – original draft:** Inge M. N. Wortel.

**Writing – review & editing:** Johannes Textor.

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
