## [Decision Letter · Decision Letter 0]

31 Oct 2022

Dear Wortel,

Thank you very much for submitting your manuscript "Constraints and trade-offs shape the evolution of T cell search strategies" for consideration at PLOS Computational Biology.

As with all papers reviewed by the journal, your manuscript was reviewed by members of the editorial board and by several independent reviewers. In light of the reviews (below this email), we would like to invite the resubmission of a significantly-revised version that takes into account the reviewers' comments. Notice that Reviewers 2 and 3 raise important concerns about the biological interpretation of your simulation results and to which extent the conclusions are supported by the results. Please address these concerns carefully in your revised submission.

We cannot make any decision about publication until we have seen the revised manuscript and your response to the reviewers' comments. Your revised manuscript is also likely to be sent to reviewers for further evaluation.

Sincerely,

Ricardo Martinez-Garcia

Academic Editor

PLOS Computational Biology

Virginia Pitzer

Section Editor

PLOS Computational Biology

Reviewer's Responses to Questions

**Comments to the Authors:**

Reviewer #1: This is an excellent paper that challenges the notion that T cell motility is evolutionarily optimized for particular goals relating to immune surveillance. Instead, the authors argue that constraints on cell movement imposed by the environment and the need to preserve membrane integrity result in patterns of movement that have previously been described as optional search strategies.

I believe this is a lovely example of how modeling can be used to simplify our thinking and pummel dogma. It emphasizes that optimality arguments are a logical and practical minefield, which - as the authors point out — has been a topic of debate in other fields, but so far not in this one. I think this paper will be an elegant contribution to Plos CB that is very much in the journal’s scope.

I only have a couple of comments.

1. (L26) I think it would be useful here to provide more detail regarding UCSP, rather than just citing ref. 13. it is an important concept and it would help to describe up-front how the basic result emerges.

2. It might be useful to highlight the potential limitations of the CP model. There is a tower of cards here - one can describe the genotype-phenotype map in terms of the CP model parameters, but this is a model itself - is it not possible that the true constraints may be entirely elsewhere? If this is the case, the utility of optimality arguments in complex biological system may truly be fruitless!

Reviewer #2: This manuscript asks an important question: should we assume that movement patterns of T cells, often considered to reflect search strategies, indicate that those search strategies have evolved, or do they simply reflect movement constraints?

The simulations show that a relationship between persistence and speed may arise due to movement constraints rather than an evolved optimization of persistence and speed to cover more area.

This result is not surprising – the USCP can be observed without invoking evolution. It is also important to consider another possibility –the T cells and the structures that constrain them may have co-evolved so that T cells find their targets within the organism’s structures efficiently. This is not necessarily the case, but the manuscript ignores that there may be co-evolution between T cells and the environments they navigate (both of course under evolutionary pressures from the host organism).

While it is logical that not all search strategies are evolvable through natural selection, it is unclear how that is relevant here. The possibility of environmental constraint determining movement patterns does not at all suggest that evolving search strategies is impossible. It is particularly misleading to suggest that the lack of speed or persistence genes hampers the ability of these traits to evolve. The genotype phenotype mapping does not require such a reductionist view. A wide range of search strategies have evolved in animal movement foraging and migration without a one-to-one mapping of genes coding for particular phenotypic traits. The broad statements about the difficulty of evolving search strategies are highly speculative and not supported by the findings of the simulation which apply only to a very specific modeling context. In general, the paper seems to overreach and draw conclusions that are not warranted by the model results.

However, the more modest claim that traits such as the USCP may be linked for genetic or physical reasons seems supported by the evidence cited.

It is warranted to stated a caveat that an observed strategy should not be declared an evolutionary optimization without careful consideration of the context and other constraints that could have led to the observation.

It is not a surprise that most biological features reflect a compromise of multiple conflicting constraints and tradeoffs so that sufficing rather than optimizing should be the expectation.

However, the question “Do we need an evolutionary explanation or is it a side effect?” is an oversimplification. The evolution of any trait is in an environmental context.

The statement that it is impossible to answer tradeoff questions in random walk models is also an overclaim. Such models can take environmental features and constraints into consideration. Not necessary to negate the possibility in order to explain

The choice of the Act- CPM model is a good one, but it is not the only possible model that can address the question of how the environment influences motion. CPM make a particular set of modeling choices to generate tradeoffs as cells move. The simulations nicely demonstrate how those tradeoffs lead to the USCP. However, the manuscript extrapolates broadly from this particular modeling choice, and that seems unwarranted.

It is also not clear that the very small number of cells and replicated in the GA are sufficient to make general claims about what can evolve more generally.

In summary, the model presents a valid criticism of prior work that tends to ignore the role of the environment in shaping T cell movement. The paper is well-written. The model is explained clearly, and the results show an existence proof that the USCP can emerge from constraints of the environment. However, this was already known in work cited in the manuscript. The manuscript should be more cautious in interpreting the results of these simulations to broader meaning without justification.

Reviewer #3: The manuscript entitled “Constraints and trade-offs shape the evolution of T cell search strategies” considers a computational model of cell crawling to study the relation between the evolution towards and optimal behavior based on a fitness depending on the maximal area covered for a single cell during a certain time. A genetic algorithm is included to modify parameter values to mimic evolution during 50 generations. Such mathematical exercise is done in order to check if evolution can trigger the different phenotypes. The methods of the study are correctly described in the manuscript and the results look interesting. However there are still open questions and comments which the authors have to address to consider the manuscript adequate for the journal

- It is not clear for me if the research done is just an interesting mathematical exercise or if the conclusions can be extrapolated to the reality of living cells. I mean I understand that the authors perform simulations to show that it is possible and it implies that it may be also possible in living cells, however this is not a proof, and evolution may affect quite different in this case. The arguments in the manuscript are in my opinion still not clear.

- The virtual cells maximizes during the evolution process described in the manuscript the ares covered, however there is a lack of explanation of the particular situation where all the trajectories in the fitness space converge. I have understood there is a compromise between increase of the persistence of the motion and to reduce cell breakup.

- On the other hand the motion is associated to a particular cell genotype, however a single cell can adopt different strategies depending on the conditions. For example, depending on the starving level, Dictyostelium discoideum cells present different persistence and speed of the motion. Therefore it is not clear for me that the genetic information determines cell speed and persistence in T cells.

- I wonder which is the area covered by a single cell for the different conditions. In figure 2 the scale of colors shows the fitness , however is it the area covered by the cell? It is proportional?

- Resulting optimized parameters depend on the original condition? I have summarized the results form the manuscript:

skin conditions (large lambda initial) max_act = 55, lambda_act = 1190

free conditions (small lambda initial) max_act = 50, lambda_act = 1165

skin conditions (small lambda initial) max_act = 50, lambda_act = 1190

In the skin conditions changing the initial value of lambda gives rise to “very similar parameters” corresponding to the same lambda_act = 1190, and (slightly?) different value of max_act, however it is the same that in the case of free conditions. This is puzzling. Is the process optimizing the area covered or just increasing speed and persistence avoiding cell break up?

**Have the authors made all data and (if applicable) computational code underlying the findings in their manuscript fully available?**

Reviewer #1: Yes

Reviewer #2: Yes

Reviewer #3: Yes

PLOS authors have the option to publish the peer review history of their article (what does this mean?). If published, this will include your full peer review and any attached files.

Reviewer #1: No

Reviewer #2: No

Reviewer #3: No
---

## [Decision Letter · Decision Letter 1]

3 Feb 2023

Dear Wortel,

We are pleased to inform you that your manuscript 'Interpreting T-cell search "strategies" in the light of evolution under constraints' has been provisionally accepted for publication in PLOS Computational Biology.

Best regards,

Ricardo Martinez-Garcia

Academic Editor

PLOS Computational Biology

Virginia Pitzer

Section Editor

PLOS Computational Biology

Reviewer's Responses to Questions

**Comments to the Authors:**

Reviewer #1: I'm satisfied with the responses.

Reviewer #2: I thank the authors for thoroughly and thoughfully addressing my previous comments. I believe the revised paper should be accepted for publication.

Reviewer #3: The authors have responded all my comments, and they have modified the manuscript. The actual version of the manuscript is more understandable than the first one, and the limitations of the model employed in their research more clear. I think the manuscript can be published in PLoS Comp Bio.

**Have the authors made all data and (if applicable) computational code underlying the findings in their manuscript fully available?**

Reviewer #1: Yes

Reviewer #2: Yes

Reviewer #3: None

PLOS authors have the option to publish the peer review history of their article (what does this mean?). If published, this will include your full peer review and any attached files.

Reviewer #1: No

Reviewer #2: No

Reviewer #3: No

---

## [Editor Report · Acceptance letter]

23 Feb 2023

PCOMPBIOL-D-22-01184R1 

Interpreting T-cell search "strategies" in the light of evolution under constraints

Dear Dr Wortel,

I am pleased to inform you that your manuscript has been formally accepted for publication in PLOS Computational Biology. Your manuscript is now with our production department and you will be notified of the publication date in due course.

With kind regards,

Anita Estes
